# Target Heart Rate Formulas for Exercise Stress Testing: What Is the Evidence?

**DOI:** 10.3390/jcm13185562

**Published:** 2024-09-19

**Authors:** Omar Almaadawy, Barry F. Uretsky, Chayakrit Krittanawong, Yochai Birnbaum

**Affiliations:** 1Department of Internal Medicine, MedStar Health, Baltimore, MD 21218, USA; omar.a.almaadawy@medstar.net; 2Division of Cardiovascular Medicine, University of Arkansas for Medical Sciences, Little Rock, AR 72205, USA; buretsky@gmail.com; 3Department of Medicine, Section of Cardiology, Central Arkansas Veterans Affairs Healthcare System, Little Rock, AR 72205, USA; 4Cardiology Division, NYU Langone Health and NYU School of Medicine, New York, NY 10016, USA; charleskritmd@gmail.com; 5The Section of Cardiology, Baylor College of Medicine, 7200 Cambridge Street, Houston, TX 77030, USA

**Keywords:** chest pain, exercise stress test, age-predicted maximum heart rate, APMHR, HRmax formula, 220-age, cardiovascular prognosis, heart rate reserve, maximum rate pressure product, metabolic equivalents, cardiovascular disease, functional status, nuclear imaging stress test, stress echocardiogram

## Abstract

Exercise stress testing (EST) is commonly used to evaluate chest pain, with some labs using 85% of age-predicted maximum heart rate (APMHR) as an endpoint for EST. The APMHR is often calculated using the formula 220-age. However, the accuracy of this formula and 85% APMHR as an endpoint may be questioned. Moreover, failing to reach 85% APMHR (known as chronotropic insufficiency) may also indicate poor cardiovascular prognosis, but measurements, such as percentage heart rate reserve (%HRR), maximum rate pressure product (MRPP), and the maximum metabolic equivalent of tasks (METs) reached during EST may provide better prediction of cardiovascular outcomes than not reaching 85% of APMHR. There is a need to incorporate comprehensive measurements to improve the diagnostic and prognostic capabilities of EST.

## 1. Introduction

Chest pain is a frequent presentation in the emergency department and in the outpatient setting, affecting annually nearly 20.5 million Americans aged >20 years [1]. It causes a significant strain on the healthcare system, with an estimated annual cost of USD 10 billion [2]. Furthermore, coronary artery disease (CAD) is responsible for 20% of deaths in adults <65 years old [3].

A standard, non-invasive, low-risk, and economical screening technique for the identification and assessment of CAD is exercise stress testing (EST) [4,5], which employs heart rate (HR), blood pressure (BP), electrocardiographic findings (ECG), and symptoms in its assessment [6].

EST is usually recommended in patients with chest pain and dyspnea who have an intermediate pretest probability for obstructive CAD [7]. EST first emerged in 1931 as a standard diagnostic method [8]. The original exercise test (“two step test”) had the patient walk up and down a short flight of stairs while monitoring the ECG, allowing ischemic ST changes to be identified if developed [8]. Subsequently, Bruce and colleagues in 1963 developed a standardized multi-stage treadmill test. Graded exercise continued until the participant could no longer proceed due to fatigue or other symptoms such as chest pain or dyspnea [9]. Although heart rate was monitored for safety reasons, achieving a specific heart rate was not the defining factor for concluding the test [9]. Subsequently, a prognostic treadmill score was developed (Duke Treadmill Score), which has been well validated [10]. The Duke Treadmill Score incorporates the development of angina, maximal exercise capacity, and ECG alterations but does not include exercise heart rate [10]. Clinicians began using 85% of age-predicted maximum heart rate (APMHR) as an endpoint for EST to indicate sufficient workload despite the lack of solid evidence supporting this practice. Furthermore, the method used to calculate APMHR may be largely inaccurate, which further compromises the validity of using 85% of APMHR as an endpoint [11,12,13,14].

This review includes the following aims:Understanding the calculation of APMHR and evaluating the major formulas currently utilized;Understanding the importance of APMHR during EST;Understanding different factors affecting APMHR;Assessing the evidence of using 85% of APMHR as an endpoint for stress testing;Understanding APMHR applications in sports medicine;Assessing other possible endpoints for EST, such as heart rate reserve and maximum rate pressure product;Understanding the metabolic equivalent of tasks (METs) and 85% of APMHR use in CAD risk assessment during EST.

## 2. Analysis of Maximum Heart Rate and Influencing Factors

### 2.1. Indications for Stress Testing

While EST is generally deemed a safe procedure, there have been reports of severe adverse events, including myocardial infarction (MI) and even death, at an approximate rate of one in every 2500 tests (0.04%). However, this was based on old data, and recent studies showed almost no major complications occurring with EST, with the most frequent complication in one study being arrhythmias, most commonly supraventricular extrasystoles [15]. Therefore, an informed clinical decision is crucial when determining the appropriateness of EST [4].

The American College of Cardiology (ACC)/American Heart Association (AHA) guidelines provide recommendations for exercise stress testing [16]. EST may be considered for patients with symptoms of typical or atypical angina pectoris who have low or intermediate pretest probability [17]. EST may also be useful in patients with established CAD who develop a change in symptoms or functional status despite medical therapy [16]. Additionally, EST may be performed in men over 40, and women over 50 who lead a sedentary lifestyle but plan to initiate vigorous exercise routines should be evaluated [18]. The recent guidelines from the European Society of Cardiology prefer stress imaging modalities over ECG EST. ECG EST can still be used to assess exercise tolerance and hemodynamic response and evaluate arrhythmias [17].

### 2.2. Maximal Heart Rate (HRmax) Definition and Importance

The maximum heart rate (HRmax) achieved during activities has clinical significance for EST [19] and is commonly used to plan exercise training or predict test endpoints. It has also been used in evaluating the extent of exertion during these tests [20,21,22].

HRmax is defined as the peak heart rate achieved during a maximal-effort exercise test, often characterized by a heart rate plateau despite a workload increase [23]. During an EST, increasing workloads necessitate an increase in cardiac output, with HR being a crucial factor in raising the cardiac output. The heart rate escalates linearly with work rate and oxygen uptake [5]. An increase in stroke volume contributes less to the exercise-induced augmented cardiac output. Unlike systolic blood pressure (SBP), which generally rises with age by ≈7 mmHg per decade, the kinetics of maximum HR and maximal oxygen consumption (VO_2_max) typically show a decline as age advances; hence, older individuals will have lower HRmax [5,24].

### 2.3. HRmax as Part of EST

For EST to be diagnostic, it is essential for patients to exert themselves to their maximal capacity, as submaximal exertion may lead to false negative results [6]. Therefore, evaluating the level of exertion during EST is crucial. The current gold standard for deciding if an individual has reached maximal exertion during exercise or activity is the peak respiratory exchange ratio (RER) of more than 1.10, which represents the total CO_2_ produced divided by the total O_2_ consumed (VO_2_), signifying true VO_2_max (100% of VO_2_) attainment [6,25]. This metric, however, is not part of routine EST as it requires special equipment to measure VO_2_ consumption and CO_2_ production. HR, on the other hand, is feasible for measuring exertion levels during exercise. A linear relationship exists between HRmax percentiles and VO_2_max percentiles [26]. Moreover, HRmax achieved is frequently used to determine exercise intensity due to its close association with VO_2_max, particularly in the 50–90% VO_2_max range, with HRmax at 90% corresponding to a VO_2_max of 85% [26,27]. Also, there is a linear relationship between HRmax and maximal exercise workload (expressed as metabolic equivalents or METs), i.e., increasing METs activity leads to increased HR [28].

However, in contrast to the linear correlation between HR and VO_2_, the relationship between HRmax and RER is not consistent. In a study involving 238 patients with symptoms of myocardial ischemia who underwent EST with additional ventilatory expired gas analysis, only approximately 50% of those achieving 85% of HRmax had an RER > 1.10 [6], i.e., maximal effort by RER had not been achieved despite reaching 85% of HRmax. This inconsistency might stem from the current equations predicting HRmax being inaccurate and other factors such as age, sex, and fitness levels influencing achievable HRmax. This topic will be explored further in the subsequent section. Since direct measurement of maximum exercise intensity (e.g., via measuring RER during EST) is usually not performed during EST, researchers, clinicians, fitness trainers, and exercise practitioners usually resort to age-based prediction equations to estimate maximum HR (age-predicted maximum heart rate or APMHR). The most frequently utilized equation is by Fox et al. of 220-age [29]. This formula has many assumptions that could lead to inaccuracy, as discussed in the upcoming sections.

### 2.4. Overview of the History of Maximum HR Prediction

The Robinson equation (1938) is one of the earliest known formulas, estimating APMHR as 212 − 0.77 × age [27,30]. This equation has a high standard error of estimate, ranging from 7 to 12 bpm [21,27]. For precise prediction of VO_2_ max based on APMHR, the prediction error should be ≤±3 bpm [27]. The Fox equation (220-age) tends to overestimate APMHR in females; hence, some researchers advocated for sex-specific formulas [31]. Furthermore, since achievable HRmax declines with age in both genders, the Fox equation tends to overestimate the APMHR in young adults and underestimate it in older adults, notably after the age of 40 [32], which can be explained as it was derived from a cohort that mainly included men under the age of 55 [33]. Another formula by Tanaka et al. derived from 351 studies involving 18,712 participants is 208 − 0.7 × age. It provides a more accurate average APMHR estimation, especially for older adults. Nevertheless, its standard error of estimate is around 11.4 bpm, and the included subjects were healthy and free of CAD, which could limit applicability to patients with CAD [11,21]. Several other formulas have been proposed, but detailing them is beyond this review’s scope (please refer to Table 1). It is noteworthy that HRmax can vary by 20–30 bpm among individuals of the same age based on equation used, so an age-driven equation may not be optimal [28].

In conclusion, the literature on HRmax determination displays considerable variation and inconsistencies. Existing methods offer a rough approximation of HRmax rather than a precise value.

### 2.5. Factors Influencing HRmax

Other than APMHR calculation via different formulas, many factors influence HR and APMHR prediction, such as sex, physical activity, smoking habits, and prescribed medications [11,12,13,14].

One of the most influential factors affecting achievable HRmax is age, which accounts for 70–80% of HRmax variability across different age groups [11,37]. The rate of decline in achievable HRmax differs significantly across various age groups, being lower in the younger population and higher in the older population [36,38]. These findings are independent of sex and physical activity [11,39]. Physiologically, this is attributed to changes in pacemaker tissue, leading to a reduction in heart reflex responsiveness, decreased adrenergic receptor sensitivity, and reduced intrinsic sinus node pacemaker activity [40,41].

Sex also plays a role in HRmax prediction. HRmax tends to be higher in young men than in women [42]. Furthermore, while achievable HRmax declines with age in both genders, this decline is more gradual in women than in men [42].

The age-predicted HRmax equations’ validity, particularly in a sample containing a sufficient number of older individuals (60 years and above), has yet to be established [35]. This is a crucial limitation, given that older adults, who have a higher prevalence of cardiovascular and other chronic conditions, are often the ones undergoing diagnostic exercise testing. Research suggests that a rigid cutoff at 85% of the APMHR does not accurately reflect the level of exertion [28].

Another essential factor is individual physical conditioning. A study found that HRmax is lower in athletes compared with those with a sedentary lifestyle [12], a finding echoed, to a lesser extent, in another study [23].

Smoking likely attenuates the achievable HRmax and induces chronotropic incompetence [13]. However, the rate of achievable HRmax decline with age is not more pronounced in smokers than in nonsmokers.

The use of drugs during exercise testing can affect HRmax. Beta-blockers and ivabradine reduce HRmax by approximately 10–15 bpm [14,43]. Therefore, the upper limit for HRmax should probably be adjusted in individuals on such medication [14]. Moreover, since not achieving 85% of APMHR during EST may carry a poor prognosis (as discussed below), the use of such medicines should be considered. Guidelines are not consistent regarding the continuation/discontinuation of beta-blockers for EST, given the fact that sudden discontinuation can lead to a rebound phenomenon, leading to an increase in HR and blood pressure [44].

Given so many factors influencing achievable HRmax, it is important to consider that our current definition and usage of HRmax/APMHR may not be accurate. In testing everyone, it is important to consider age, sex, prior medical history, fitness level, and pharmacologic history to assess their HR and exercise capacity better.

## 3. The Use of 85% of the Age-Adjusted HR as a Cutoff for Stress Tests

### 3.1. Historical Context and Current Practice

Since the introduction of EST, 85% of the APMHR (calculated using the formula 220-age) has been utilized as an endpoint for the test. Moreover, it was found that 40% of laboratories consider 85% of APMHR as the primary exercise endpoint [28]. However, the evidence for using this endpoint (85% of APMHR) during EST is lacking.

### 3.2. HR and Sports Medicine

EST is commonly administered to asymptomatic, healthy individuals to help uncover hidden health conditions, mitigate exercise-related risks, and evaluate physical performance. Accurate maximum HR values are essential in defining training loads, assessing exercise intensity (via creating training zones based on maximum HR), and evaluating the impact of these training regimens [45]. Furthermore, the percentage of APMHR achieved during exercise correlates with VO_2_max, which is a measure of cardiorespiratory fitness, as 55%, 70%, 85%, and 90% of the APMHR, which correspond to 40%, 60%, 80%, and 90% of VO_2_max [46]. The American College of Sports Medicine (ACSM) recommends utilizing a percentage of HRmax to measure exercise intensity, which is critical to enhancing exercise endurance, and provides recommendations about the optimal exercise duration and intensity to gain various health benefits. Various guidelines have been established based on the percentage of the maximum HR achieved. For example, reaching 64–76% of APMHR would be considered moderate-intensity exercise, while vigorous exercise is 77–95% of APMHR [47,48].

However, data are conflicting, and reaching 85% of APMHR may not be appropriate as it may not reflect the actual level of exertion. A cross-sectional study that included 469 healthy adults following a supervised exercise program (2–3 sessions/week) demonstrated that using 85% of APMHR as a cutoff underestimated exercise capability [49]. When participants were allowed to exercise to their maximum capacity, most of them were able to achieve >85% of APMHR [49]. This observation is consistent with findings from other studies, including the study conducted by Jain et al., suggesting that a reevaluation of the 85% APMHR benchmark may be warranted to more accurately assess individual exercise capacity and the prevalence of inducible ischemic events [28,49]. Moreover, as discussed above, for EST to be diagnostic for CAD, it is essential for patients to exert themselves to their maximal capacity, as submaximal exertion may lead to false negative results [6].

### 3.3. Debate Surrounding Using 85% of HRmax as an Endpoint for EST

Using 85% of the APMHR as an endpoint for EST as an indication of maximal exercise intensity is likely arbitrary. Overall, achieving or not achieving 85% of HRmax does not consistently correlate with determining exercise capacity. Its continued use is a matter of debate among cardiologists and exercise physiologists. Given the conflicting evidence supporting this endpoint, other endpoints must be considered.

It is assumed that reaching 85% of the APMHR during EST indicates reaching maximal exercise capacity. However, in one study that focused on patients undergoing stress tests for diagnosing CAD or evaluating patients with known CAD, most participants could exercise on average for an additional 3.5 min after reaching 85%APMHR, calculated by the formula 220-age [28]. When comparing their ECG at 85% of APMHR and at peak (100%) exercise, the ST segment depression measured 1.2 ± 0.7 mm and 2.3 ± 0.9 mm, respectively (*p* < 0.001) [28]. The authors suggested that peak exercise rather than 85% APMHR might be a more appropriate EST endpoint. Peak exercise leads to more pronounced ST segment deviation in this study, which influenced the Duke Treadmill Score and, therefore, the prediction of CAD. It should be noted that a significant portion (79–87%) of this cohort did not have confirmed CAD. Most of the individuals had a follow-up single-photon emission-computed tomography (SPECT) study, which was abnormal for 69% of the individuals in the group who did not achieve 85% of APMHR. It was only abnormal in 28% of the individuals who exercised for >3 min after reaching 85% of APMHR, which was attributed to post-test bias; however, a false positive test result cannot be totally excluded. Thus, the accuracy (false positive rate) of a more intense exercise versus a goal of 85% of the APMHR is unclear. Therefore, the degree of ST segment depression should not be used as a marker for the extent of induced ischemia during EST.

On the other hand, failing to reach 85% of APMHR during EST, which may be classified as “chronotropic insufficiency”, a term that was first prescribed in 1975 for individuals who did not have appropriate heart rate augmentation in response to exercise [50], can aid in interpreting and identifying patients at risk of CAD. The Framingham Heart Study indicated that individuals with chronotropic insufficiency faced an elevated risk of developing coronary heart disease and premature death [51]. A Cleveland Clinic study revealed that adults not on β blockers and referred for symptom-limited treadmill thallium testing had an increased mortality risk if they did not achieve 85%APMHR during EST (HR: 1.84; 95% CI: 1.13 to 3.00) [52]. Another study found that not reaching 85%APMHR was linked with increased mortality (adjusted risk [AR], 1.49; 95% CI, 1.02 to 2.22; *p* = 0.04) and cardiac death (AR, 2.13; 95% CI, 1.10 to 4.17; *p* = 0.03) in a cohort of 3221 patients aged 12–59 years undergoing treadmill exercise echocardiography [53]. However, patients who did not reach 85% APMHR had a greater incidence of prior MI, complicating study interpretation [53].

It has also been found that failing to reach 85% APMHR during EST increased the risk of sudden death by 80% (*p* = 0.001), cardiac death by 40% (*p* < 0.001), and all-cause death by 30% (*p* < 0.001). The hazard ratios for these risks were 1.8, 1.4, and 1.3, respectively [54]. Notably, for those for whom EST was terminated due to ST segment abnormalities, there was a much higher risk of sudden death (hazard ratio 5.0, 95% CI 3.0 to 8.4) and all-cause mortality (hazard ratio 1.9, 95% CI 1.7 to 2.2) [54], supporting the concept that ischemia induced at lower levels of exercise is associated with increased adverse events from CAD. Thus, the reason for the inability to achieve the target heart rate (induced ischemia, deconditioning, drug effects, etc.) could affect this association.

To summarize, not reaching 85% APMHR during EST is associated with a poorer prognosis and risk of developing future adverse coronary events. This finding, however, has low sensitivity. It was found that achieving >85% of APMHR yielded a sensitivity of 60% and a specificity of 53% for the absence of cardiovascular events over an average follow-up of 5 years, compared to a sensitivity of 95% and a specificity of 25% if 95% of APMHR is reached. This makes it an unsatisfactory endpoint for EST while reaching maximal rate pressure product (MRPP) of >25,000 has a sensitivity of 97% for the absence of cardiovascular events over an average follow-up of 5 years [55] (please see below). The MRPP study was primarily conducted on patients without known heart disease, suggesting that the results might be most applicable to new patients referred to EST for CAD diagnosis. Also, it shows that reaching 85% HRmax during EST may be beneficial prognostically, but there may still be better parameters to monitor, such as MRPP [55].

Contrary to myocardial perfusion imaging, where the isotope is injected during maximal stress to evaluate perfusion defects, stress echocardiography looks for indirect signs of ischemia, particularly regional wall motion abnormalities. The American Society of Echocardiography recommends obtaining imaging within 1–2 min after exercise, with some authors suggesting 60 s, as wall motion abnormalities can resolve quickly after exercise termination [56]. An animal study that included 10 mongrel dogs studied the effect of exercise duration and intensity on postischemic myocardial dysfunction and found that high-intensity exercise compared to 50% low-intensity exercise led to greater degrees of myocardial dysfunction after exercise. Specifically, during high-intensity exercise with coronary artery stenosis, there was a higher degree of myocardial dysfunction, with wall thickening being only 44 ± 23%, less than the thickness observed in the lower intensity group of 60 ± 18% (*p* < 0.01) [57]. It was also found that imaging at peak exercise (by bicycle stress echocardiogram) and post-stress after a treadmill stress test showed that the sensitivity of peak (bicycle) versus post-stress (treadmill) was 82% and 75%, with a specificity of 80% and 90%, respectively. This translates into better ischemia detection [58]. Hence, during stress echocardiography, the goal should be reaching peak exercise intensity and duration, regardless of the achieved maximum HR.

In conclusion, setting 85% of APMHR as an endpoint for EST may not be ideal, and test performers should aim for peak exercise intensity. Moreover, incorporating scores such as the Duke Treadmill Score, which incorporates exercise time, ECG changes, and symptoms, may be better diagnostic- and prognostic-wise than relying only on ECG changes [59].

## 4. Other Endpoints That Are Based on HRmax

As mentioned above, reaching 85% APMHR carries significant prognostic value. However, other parameters might be more useful and sensitive in predicting cardiac events. These may include the percentage of heart rate reserve (%HRR)—[maximum HR -resting HR/(220-age)—resting heart rate]—and the maximum rate pressure product (MRPP), which is the product of HR and systolic blood pressure.

The inability to achieve >80% of the %HRR during EST is considered abnormal, and it was more effective than the failure to achieve 85% of the APMHR in identifying and stratifying patients at an increased risk of CAD [60]. It was observed that patients who underwent the SPECT study and had a moderate summed stress score (score used in SPECT to assess the extent and severity of combined ischemia and infarction in the left ventricle of the heart), coupled with a normal %HRR (>80%), exhibited less than a 1% rate of cardiovascular event and death during the study’s follow-up period of median 677 days [60]. In contrast, those who achieved 85% of their APMHR, along with a moderate summed stress score, exhibited a >1% rate of cardiovascular events and death greater than 1% [60]. Overall, using the %HRR led to the identification of 2.2 times more individuals at higher risk for cardiac death, compared to the inability to achieve 85% APMHR [60]. This research suggests that calculating %HRR could provide additional prognostic information, which might influence further assessment and management of these patients [60]. Another study [61] showed that the optimal cut point for the %HRR is 95.9%, which showed 81% sensitivity for the prediction of future cardiovascular events; however, when compared to reaching 95% of APMHR, it was statistically insignificant. Hence, data regarding %HRR are conflicting, and they should be incorporated with other parameters when used as a prognostication tool. Reaching MRPP >25,000 had a sensitivity and specificity of 97% and 45%, respectively, for predicting the absence of cardiovascular events over a mean period of 5 years, while achieving 85% APMHR had a sensitivity and specificity of only 60% and 53%, respectively [55]. This small study included only 236 patients referred to EST due to chest pain with no known CAD who had inconclusive EST due to poor functional capacity (defined by a functional capacity < 85% of age and sex, measured by achieving <85% of the appropriate METs for age and sex). In a 2-year follow-up, no obstructive CAD was found in MRPP > 25,000 patients, while 31% of MRPP < 25,000 patients had obstructive CAD [55]. A similar study, by the same authors, performed in a cohort of 1080 participants determined that MRPP of 25,085 reached during stress echocardiogram had a sensitivity and specificity of 75.2% and 75.6%, respectively, in predicting the absence of cardiovascular events [61]. The discrepancy in sensitivity between the two studies may stem from differing modalities and baseline characteristics. The previous study used EST and excluded patients with CAD, while the study used stress echocardiogram and included patients with CAD.

Similar findings were noted in another analysis where patients with negative EST results (i.e., no ST segment depression on ECG) were examined [62]. It was found that the predictive value for the absence of CAD increased from 60% to 86% in those patients who achieved MRPP > 30,000. Furthermore, the predictive value for the absence of multivessel CAD rose from 81% to 100%.

During exercise, the average systolic blood pressure (SBP) rise is around 8–12 mmHg/MET, with 190–220 mm Hg as the maximum. A clinical recommendation for systolic blood pressure is that it should not exceed 250 mm Hg during EST [5]. However, patients with aortic stenosis, left ventricular outflow obstruction, and bilateral subclavian artery stenosis may have lower measured blood pressure [63,64], which may lead to lower MRPP and hence limit its utility.

Given the above data, we recommend incorporating MRPP into the interpretation of exercise stress testing (EST). This is particularly relevant for patients with no known CAD who cannot reach 85% APMHR and do not display ischemic ST changes. In these scenarios, additional investigation might be unnecessary if the patient achieves an MRPP of 25,000–30,000. Nonetheless, this should be clinically evaluated, considering other factors such as pretest probabilities and patient risk factors.

## 5. HRmax, Stress Echocardiogram, and Nuclear Imaging Studies

Unlike regular exercise stress testing (EST), which only uses an ECG to detect ischemia, imaging stress tests offer superior diagnostic capabilities [65]. Stress echocardiograms, for instance, provide additional information about the heart’s structure and function [65].

Stress echocardiography is a safe and feasible study with high diagnostic accuracy for CAD, with serious adverse events reported in about 1 in 6574 cases [66,67]. It is superior to ECG EST and offers comparable results to SPECT [56,67,68], with diagnostic accuracy reaching up to 90% [67]. Stress echocardiography incorporates various findings that aid in CAD diagnosis and mortality risk assessment, including regional wall motion abnormalities (RWMAs) and coronary flow velocity reserve (CFVR), which can be assessed using Doppler imaging during semi-supine exercise [67]. Additionally, heart rate reserve, calculated as peak HR/rest HR, can also be measured during stress echocardiography [67,69]. The ABCDE score applied during the procedure helps determine annual all-cause mortality risk, ranging from 0.4% (score 0) to 2.7% (score 5) [67]. The protocol includes assessing wall motion abnormalities (A), lung ultrasound for B lines (B), left ventricular contractile reserve (C), CFVR in the LAD (D), and heart rate reserve (E) [69]. A heart rate reserve is considered positive if it is less than 1.80 during an exercise stress echocardiogram, with an HR of 2.955 (95% CI 1.848–4.725; *p* < 0.001) for all-cause mortality, during 21-month follow-up, when positive [69]. Assessment of left ventricular volumes, such as end-systolic volume (used in calculating left ventricular contractile reserve) and end-diastolic volumes, can also be performed; however, they are not routinely performed as they are time-intensive and may require a contrast agent for better imaging [67].

Nuclear myocardial perfusion stress imaging (MPI) uses radiological tracers and is highly sensitive in detecting single-vessel disease. MPI is also the best choice for patients with conditions like left bundle branch block or those with pacemakers that cause abnormal septal motion [65].

Reaching 85% of APMHR is not limited to EST; it also applies to nuclear imaging studies. In patients with CAD, performing intense, symptom-limited exercise followed by milder exercise at 70% APMHR, it was found that the size and extent of the ischemic area on Thallium-201 myocardial imaging decreased at milder exercise compared to symptom-limited EST [70]. Moreover, it was also found that exercise SPECT thallium imaging was considerably better in detecting ischemia in patients who reached ≥85% APMHR compared with those at submaximal heart rates. In a study of 330 patients who underwent coronary angiography and exercised SPECT thallium-201 imaging within 6 months, two groups were compared: one who exercised to ≥85% APMHR and the other with <85% APMHR. The group that reached ≥85% APMHR demonstrated a higher frequency of perfusion defects compared to those who achieved submaximal HR. SPECT thallium detected perfusion defects in 74%, 88%, and 98% of patients with one-, two-, and three-vessel CAD, respectively, in the target HR group. This contrasted with 52%, 84%, and 79%, respectively, in the submaximal HR group (*p* < 0.05) [71]. Hence, adequate exercise is essential for ischemia detection.

It is crucial to acknowledge that radiation exposure has numerous side effects, including damage to the skin, hair, and eye lens, as well as the potential for germ cell mutations and cancer development [69]. Contrary to the belief that only higher doses lead to cancer, there is no threshold level below which cancer cannot occur [72], meaning even small doses carry a risk. Stress echocardiograms with contrast have been found to be as accurate as 99-technetium SPECT in diagnosing coronary artery disease [73]; hence, it may be a better option to reduce the risk of radiation exposure.

The American Society of Nuclear Cardiology recommends using a symptom-limited test. It was even found that in patients with coronary heart disease (especially in patients taking medications such as β-blockers), the test preserves its prognostic value without achieving 85% APMHR. This emphasizes reaching peak exercise guided by symptoms and that reaching the highest HR possible should be the aim/termination point of the nuclear imaging study instead of reaching a predetermined HR.

## 6. HRmax and METs

EST employs metabolic equivalents (METs) to evaluate exercise capacity. One MET is quantified as 3.5 mL O_2_ uptake/kg/min. In the St. James Women Take Heart Project, it was found that women referred for EST to assess for CAD who attained METs of less than 4 had a statistically significant 89% and 121% higher risk of death from any cause and cardiac causes, respectively [74]. Furthermore, attaining high MET levels (>10) during EST correlates with a favorable prognosis and a reduced risk of developing symptomatic CAD [4].

However, achieving >10 METs alone does not guarantee a favorable prognosis, underscoring the need to consider the maximum HR attained. Optimal prognostic indications are typically associated with subjects reaching both >10 METs and 85% of their APMHR (calculated using 220-age). A study of patients with intermediate to high clinical risk of CAD who underwent both exercise testing and SPECT imaging demonstrated that among those who achieved ≥10 METs with no ischemic ST depression, only 0.7% manifested 5–9% myocardial ischemia, with no cases of ≥10% of the left ventricle involvement. Conversely, of the patients who achieved ≥10 METs with ischemic ST depression, 2 (4.7%) exhibited 5–9% of the left ventricle myocardial ischemia, and 2 (4.7%) demonstrated ≥10% myocardial ischemia (*p* = 0.016 and <0.001, respectively). The study observed a significantly higher incidence of ischemia by SPECT in patients unable to attain ≥10 METs. Individuals achieving ≥10 METs had a five-times lower incidence of reversible ischemic defects and 2.6 times fewer fixed perfusion defects compared to those attaining <7 METs, denoting a poor workload [75].

Subjects achieving ≥10 METs but not achieving 85% of APMHR were approximately four times more likely to exhibit fixed and ischemic perfusion defects compared to those reaching both ≥10 METs and ≥85% of their APMHR. However, this subgroup, i.e., achieving <85% of APMHR, displayed higher rates of β-blocker use on the day of testing and a 2.2-fold higher prevalence of known CAD compared with patients who reached ≥85% of APMHR and ≥10 METs; thus, the association is confounded by the different baseline parameters [74].

Failure to achieve either <10 METs or <85% of APMHR is recognized as a risk factor for both the presence of CAD and subsequent cardiac events [53,75].

These studies underscore the importance of using 85% of APMHR as a prognostic tool and considering METS achieved in considering prognosis.

## 7. Conclusions

Since its inception in 1931, EST has undergone numerous enhancements, focusing on the integral metric of HRmax and other parameters such as MRPP and METs in assessing adequate workload and stress on the heart. Many centers are still using 85% of APMHR as an endpoint for EST. However, many factors influence HRmax, especially age, which accounts for 70–80% of its variability, with sex also playing a considerable role.

Reaching 85% of APMHR during EST may have some predictive value; however, there are other superior parameters.

In conclusion, there remains a need for refined methodologies in APMHR determination, as they are crucial for improving the effectiveness of EST in the assessment and management of cardiovascular diseases. Achieving advancements in this field may utilize combined approaches, including HRmax, MRPP, and METs, to interpret the EST results.

## Figures and Tables

**Table 1 jcm-13-05562-t001:** Different formulas used in calculation of APMHR.

Author	HRmax Equation	Comment
Fox’s formula [29]	220-age	Was obtained from a cohort of men with an average age of 55 years
Londeree and Moeschberger [34]	206 − 0.7 × age	Aligns with the Tanaka formula
Fairbarn [35]	208 − 0.8 × age	For males
Robergs and Landwehr [11,27]	208.754 − 0.734 × age	Aligns with the Tanaka formula
Nes’s formula [23]	211 − 0.64 × age	Was examined in healthy men and women
Lach et al. complex multivariate model [36]	229 − 0.64 × age − 0.23 × body mass + 0.02 × BMI − 0.38 × VO_2_max + 0.33 × body fat + 0.02 × fitness level + 8.74 × sex + 0.97 × testing modality	Includes additional variables such as body weight, VO_2_max, sex, and test type and has demonstrated the lowest error rate when estimating HRmax

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
