# Peer review of "Target Heart Rate Formulas for Exercise Stress Testing: What Is the Evidence?"

_jcm, 2024, doi:10.3390/jcm13185562_

Round 1

Reviewer 1 Report

Comments and Suggestions for Authors

This comprehensive review well analyses the influence of achieved heart rate and of other parameters in the productivity of a stress test.

The review is clear and useful in evaluating the results of a stress test.

According to the text, a failure to achieve an 85% maximum heart rate has a negative prognostic value. However, this failure may also be due to beta-blockers use. A paragraph analysing the influence of pharmacologic bradycardia on the sensitivity and specificity of the stress test would be useful.

On line 78 "reports of i severe" has an extra i

Author Response

Comment 1: According to the text, a failure to achieve an 85% maximum heart rate has a negative prognostic value. However, this failure may also be due to beta-blockers use. A paragraph analysing the influence of pharmacologic bradycardia on the sensitivity and specificity of the stress test would be useful.

Response 1:

Thanks for pointing this out, we agree with you. We added “Moreover, since not achieving 85% of APMHR during EST may carry a poor prognosis (as discussed below), the use of such medicines should be considered. Guidelines are not consistent regarding continuation/discontinuation of beta-blockers for EST, given the fact that sudden discontinuation can lead to a rebound phenomenon leading to increase in the HR and blood pressure [66]“ in the lines 194-198.

Comment 2: On line 78 "reports of i severe" has an extra i

Response 2: Thanks for pointing this out. We edited the manuscript and removed the extra “i” in line 80.

Reviewer 2 Report

Comments and Suggestions for Authors

2024 jcm-3139819, Almaadawy O. et al.,

The authors focused on, in their review manuscript, the importance of heart beats (heart rate) related to exercise stress test (EST), which commonly used to the evaluation of chest pain, with the age-predicted maximum heart rate (APMHR) as an important endpoint in the EST. However, the accuracy of APMHR as a final outcome could be a subject of debate and may indicate poor cardiovascular prognosis. The authors concluded that needs remain in methodologies of APMHR determination, which are crucial for improving the effectiveness of EST in the assessment and management of various cardiovascular diseases, utilizing combined approaches, including the interpretation of HRmax, MRPP, and METs.

 Comments:

-        The manuscript is well written, however, the checking of the English (e.g., grammatical errors and typos) is not the duty of this reviewer. Some typos might be occurred throughout the manuscript. Rechecking of the manuscript may be useful.

The manuscript was evaluated on the scientific merit by this reviewer. The manuscript (jcm-3139819) is not a brand new one, however, gives a valuable picture about the importance of EST and APMHR.

Suggestions:

The heart, including heart rate right and left ventricular function, is the key organ of the blood circulation in living subjects including e.g., reptiles, fish, birds, and mammals, leading to in conjunction with an extensive network of blood vessels to supply all cells and tissues with nutrients to serve their vital and physiological function. Heart failure, ventricular arrhythmias, gene mutations and various stress factors (stressors) cause sudden cardiac deaths in human subjects reflected in action potential (AP) and ECG changes. Therefore, tools used for the prevention and/or prediction of cardiac arrhythmias, heart rate and heart failure are critical points in the human society.

Other endpoints are available on HRmax?

Is there any connection among HRmax, imaging and electrocardiogram (ECG)? Please, give an explanation.

Is EST the best in the assessment and management/prediction of cardiovascular diseases (e.g., chest pain), or the ECG itself may be alone, and/or other indices are most important for prediction? This point could be clarified in the manuscript.

An additional Figure related to the induction of cardiovascular stress or its potential factors (as stressors) may further improve the quality of the revised manuscript.

Finally, it is well known that the shape of action potential (AP) determines the regular contraction of the myocardium and the physiological activity of the heart, which reflects in ECG recordings. Changes in the heart rate lead to abnormal function of the myocardium, however, these changes may be different in animal subjects and human beings. It is not a new observation, for instance, that slow heart rate can be protective (preventing the development of arrhythmias) under experimental conditions (J Pharmacol Exp Ther. 1982 Aug;222(2):424-9. PMID: 7097563; Cardiovasc Res. 1988 Nov;22(11):818-25. doi: 10.1093/cvr/22.11.818; Am J Physiol. 1989 Jan;256(1 Pt 2):H21-31. doi: 10.1152/ajpheart.1989.256.1.H21.PMID: 2912184; Cardiovasc Res. 2004 Feb 1;61(2):208-17. doi: 0.1016/j.cardiores.2003.11.018.PMID: 14736537; Cell Rep. 2022 Mar 8;38(10):110468. doi: 10.1016/j.celrep.2022.110468.PMID: 35263588), while in human hearts the slow heart beat can be arrhythmogenic (e.g., Long QT, Brugada syndrome, and torsades de pointes arrhythmias), leading to a fatal outcome (sudden cardiac death). The difference between animal and human studies may be related to the activities and/or sensitivities of various ion channels/genes under various stress conditions (Heart Dis Stroke. 1993 Jan-Feb;2(1):75-80. PMID: 7511968; Heart Rhythm. 2005 May;2(5):507-17. doi: 10.1016/j.hrthm.2005.01.020.PMID: 15840476; Acta Physiol Scand. 2005 Dec;185(4):291-301. doi: 10.1111/j.1365-201X.2005.01496.x.PMID: 16266370; Front Pharmacol. 2020 May 12;11:616. doi: 10.3389/fphar.2020.00616. eCollection 2020. PMID: 32477118; Adv Exp Med Biol. 2024;1441:1033-1055. doi: 10.1007/978-3-031-44087-8_66.PMID: 38884768). Thus, the suggested references (above) might be interested and an important point, which could be mentioned, cited and briefly discussed in the Introduction and/or Conclusion of the revised version.

This reviewer believes that the incorporation of the aforementioned publications (classic and recent) in the revised version of the manuscript may substantially increase the interest of general readers, surgeons, senior and junior clinicians and experimental researchers.

Comments on the Quality of English Language

Checking of the English (e.g., grammatical errors and typos) is not the duty of this reviewer. Some typos might be occurred throughout the manuscript. Rechecking of the manuscript may be useful.

Author Response

Comment 1: Other endpoints are available on HRmax?

Response 1: Thanks for your comment. Most of the published articles focused on the 85% of age predicted maximum heart rate (APMHR) as a HRmax endpoint. Unfortunately, we could not find much data regarding other endpoints. Whitman et. al. In the article “Is downstream cardiac testing required in patients with reduced functional capacity and otherwise negative exercise stress test? A single center observational study” mentioned that 95% of APMHR, if reached during EST, have better sensitivity than 85% of APMHR in predicting future cardiovascular events. We edited the manuscript to address that in lines 285-286

Comment 2: Is there any connection among HRmax, imaging and electrocardiogram (ECG)? Please, give an explanation.

Response 2: Thanks for your comment. Yes, there is a connection between HRmax achieved during EST and ECG. In the lines 246-249, we mentioned how the ST segment depression changed at different HRmax achieved during the EST. Regarding the imaging, in the lines 388-391, we mentioned how achieving higher HRmax during SPECT can lead to better ischemia detection.

Comment 3: Is EST the best in the assessment and management/prediction of cardiovascular diseases (e.g., chest pain), or the ECG itself may be alone, and/or other indices are most important for prediction? This point could be clarified in the manuscript.

Response 3: Thanks for pointing this out. We added, “Moreover, incorporating scores such as Duke Treadmill Score, which incorporates exercise time, ECG changes and symptoms may be better diagnostic and prognostic wise than relaying only on ECG changes [72].” to the lines 311-314

Comment 4: An additional Figure related to the induction of cardiovascular stress or its potential factors (as stressors) may further improve the quality of the revised manuscript.

Response 4: Thanks for your comment, would you please clarify this further, and we are happy to accommodate anything that would enhance the manuscript quality.

Comment 5: Finally, it is well known that the shape of action potential (AP) determines the regular contraction of the myocardium and the physiological activity of the heart, which reflects in ECG recordings. Changes in the heart rate lead to abnormal function of the myocardium, however, these changes may be different in animal subjects and human beings. It is not a new observation, for instance, that slow heart rate can be protective (preventing the development of arrhythmias) under experimental conditions (J Pharmacol Exp Ther. 1982 Aug;222(2):424-9. PMID: 7097563; Cardiovasc Res. 1988 Nov;22(11):818-25. doi: 10.1093/cvr/22.11.818; Am J Physiol. 1989 Jan;256(1 Pt 2):H21-31. doi: 10.1152/ajpheart.1989.256.1.H21.PMID: 2912184; Cardiovasc Res. 2004 Feb 1;61(2):208-17. doi: 0.1016/j.cardiores.2003.11.018.PMID: 14736537; Cell Rep. 2022 Mar 8;38(10):110468. doi: 10.1016/j.celrep.2022.110468.PMID: 35263588),
while in human hearts the slow heart beat can be arrhythmogenic (e.g., Long QT, Brugada syndrome, and torsades de pointes arrhythmias), leading to a fatal outcome (sudden cardiac death). The difference between animal and human studies may be related to the activities and/or sensitivities of various ion channels/genes under various stress conditions (Heart Dis Stroke. 1993 Jan-Feb;2(1):75-80. PMID: 7511968; Heart Rhythm. 2005 May;2(5):507-17. doi: 10.1016/j.hrthm.2005.01.020.PMID: 15840476; Acta Physiol Scand. 2005 Dec;185(4):291-301. doi: 10.1111/j.1365-201X.2005.01496.x.PMID: 16266370; Front Pharmacol. 2020 May 12;11:616. doi: 10.3389/fphar.2020.00616. eCollection 2020. PMID: 32477118; Adv Exp Med Biol. 2024;1441:1033-1055. doi: 10.1007/978-3-031-44087-8_66.PMID: 38884768).
Thus, the suggested references (above) might be interested and an important point, which could be mentioned, cited and briefly discussed in the Introduction and/or Conclusion of the revised version.

Response 5: Thank you for your comment. While the issues you raised are important, we believe that those are out of scope for the current review and should be addressed in a separate review.

Reviewer 3 Report

Comments and Suggestions for Authors

Comments: “Target Heart Rate Formulas for Exercise Stress Testing: What Is the Evidence” is a review addressing the important issue of maximum heart rate as an endpoint for the exercise stress tests (EST). The authors presented some interesting historical facts about the different formulas of age-predicted maximum heart rate (APMHR). The paper was aimed to review the contemporary understanding the evidence for their application and comparison some different other parameters as diagnostic and prognostic tools during (EST). However, some issues need to be addressed:

Questions/Comments:

Major issues:

1.     Page 2. The authors begin by discussing the safety of EST, which is a reasonable starting point for the review. They state that the myocardial infarction and death rate is approximately 1 in 2,500 ESTs. However, this statistic is drawn from Kharabsheh SM, Al-Sugair Abdulaziz, Al-Buraiki Jehad, and Farhan J.'s "Overview of Exercise Stress Testing" published in Annals of Saudi Medicine. This review, published in 2006, was based on even older and smaller data sets. In contrast, one of the largest multicenter studies published in the same year showed a much lower major adverse cardiac event (MACE) rate. This study, which included 26,295 ESTs, reported an event rate of 1 in 6,574 cases, with no deaths during the exercise tests [1]. Moreover, recent single-center studies have shown even lower complication rates. For example, [2] a study comprising 10,250 patients reported no deaths, acute myocardial infarctions, ventricular fibrillation, or asystole, with only non-sustained ventricular tachycardia occurring in 114 cases (1.1%). Modern studies also report no severe complications during exercise, likely due in part to the use of semi-supine exercise stress echocardiography. During modern exercise echocardiography, severe ischemia is visualized earlier, often before ST-changes or symptoms appear, resulting in shorter and safer tests.

Therefore, I do not agree with the claim that the myocardial infarction and death rate is approximately 1 in 2,500 ESTs in 2024. This should be reflected in contemporary papers.

2.     Page 1-2. Introduction/Indications for stress testing and further… Several times throughout the text, the authors mention in passing that the indication for EST is as a diagnostic tool for patients with chest pain. This is an outdated and narrow approach that limits the scope of EST's potential applications. In addition to chest pain, dyspnea is a very important symptom that can also serve as an indication for EST. Furthermore, there are many other pathologies that could warrant the use of EST [3,4].

3.      Page 5. The paper compares the rate of ST-depression during exercise with different APMHR formulas. However, this comparison is of limited importance since the diagnostic sensitivity of ST-depression is not very high, as noted in the new guidelines [4]. This is why imaging-based EST is now considered the first line of assessment for most patients. The study on differences in ST-segment depression was published in 2011 and is a pilot study involving 306 patients.

4.     The paper is a review, not an original study, so modern modalities should be thoroughly described. However, there is only a single sentence in the “HRmax and Nuclear Imaging Studies” section regarding stress echocardiography. Stress echocardiography, by the way, can provide a number of new parameters with proven prognostic value. In relation to the paper's topic, heart rate reserve (HR reserve) has been proven to be an independent predictor of mortality [5]. It is a simple yet important parameter that can be used alone or in combination with other parameters, as demonstrated in the large multicenter study, SE2020.

5.     Page 8. In my opinion, the “HRmax and Nuclear Imaging Studies” paragraph should be significantly reworked. The authors analyze a 1989 study involving 330 patients, discussing the high and adequate ischemia detection using the SPECT thallium test. This might give readers the impression that this test is a good alternative for ischemia detection in current clinical practice. However, it should be emphasized that this test has the same accuracy as the older and simpler exercise stress echocardiography. Furthermore, safety concerns should also be highlighted. Since 2010, numerous studies have pointed out the significant dangers associated with SPECT tests.  For example, “Using radiation safely in cardiology: what imagers need to know. Heart. 2019: “Might best practices for SPECT myocardial perfusion imaging: 1. Avoid thallium stress…” [6].

“The appropriate and justified use of medical radiation in cardiovascular imaging: a position document of the ESC Associations of Cardiovascular Imaging, Percutaneous Cardiovascular Interventions and Electrophysiology, 2014” [7] etc., including recent papers [8]. This issue should be discussed meticulously and accurately.

The references beneath are only to prove my words, as not to be unfounded. You could not to use them, you can use other recent papers, as you feel appropriate.

1.     Varga A, Rodriguez Garcia MA, Picano E; International Stress Echo Complication Registry. Safety of stress echocardiography (from the international stress echo complication registry). Am J Cardiol. 2006;98(4):541-3.

2.     Andrade SM, Telino CJ, Sousa AC, Melo EV, Teixeira CC, Teixeira CK, Santana JS, Mota IL, Matos CJ, Oliveira JL. Low Prevalance of Major Events Adverse to Exercise Stress Echocardiography. Arq Bras Cardiol. 2016 Aug;107(2):116-23. doi: 10.5935/abc.20160096.

3.     Picano E, Pierard L, Peteiro J, Djordjevic-Dikic A, Sade LE, Cortigiani L, Van De Heyning CM, Celutkiene J, Gaibazzi N, Ciampi Q, Senior R, Neskovic AN, Henein M. The clinical use of stress echocardiography in chronic coronary syndromes and beyond coronary artery disease: a clinical consensus statement from the European Association of Cardiovascular Imaging of the ESC. Eur Heart J Cardiovasc Imaging. 2024 Jan 29;25(2):e65-e90. doi: 10.1093/ehjci/jead250.

4.     Knuuti J, Wijns W, Saraste A, Capodanno D, Barbato E, Funck-Brentano C, Prescott E, Storey RF, Deaton C, Cuisset T, Agewall S, Dickstein K, Edvardsen T, Escaned J, Gersh BJ, Svitil P, Gilard M, Hasdai D, Hatala R, Mahfoud F, Masip J, Muneretto C, Valgimigli M, Achenbach S, Bax JJ; ESC Scientific Document Group. 2019 ESC Guidelines for the diagnosis and management of chronic coronary syndromes. Eur Heart J. 2020 Jan 14;41(3):407-477. doi: 10.1093/eurheartj/ehz425. Erratum in: Eur Heart J. 2020 Nov 21;41(44):4242. doi: 10.1093/eurheartj/ehz825.

5.     Ciampi Q, Zagatina A, Cortigiani L, Wierzbowska-Drabik K, Kasprzak JD, Haberka M, Djordjevic-Dikic A, Beleslin B, Boshchenko A, Ryabova T, Gaibazzi N, Rigo F, Dodi C, Simova I, Samardjieva M, Barbieri A, Morrone D, Lorenzoni V, Prota C, Villari B, Antonini-Canterin F, Pepi M, Carpeggiani C, Pellikka PA, Picano E. Prognostic value of stress echocardiography assessed by the ABCDE protocol. Eur Heart J. 2021 Oct 1;42(37):3869-3878.

6.     Williams MC, Stewart C, Weir NW, Newby DE. Using radiation safely in cardiology: what imagers need to know. Heart. 2019 May;105(10):798-806. doi: 10.1136/heartjnl-2017-312493.

7.     Picano E, Vañó E, Rehani MM, Cuocolo A, Mont L, Bodi V, Bar O, Maccia C, Pierard L, Sicari R, Plein S, Mahrholdt H, Lancellotti P, Knuuti J, Heidbuchel H, Di Mario C, Badano LP. The appropriate and justified use of medical radiation in cardiovascular imaging: a position document of the ESC Associations of Cardiovascular Imaging, Percutaneous Cardiovascular Interventions and Electrophysiology. Eur Heart J. 2014 Mar;35(10):665-72. doi: 10.1093/eurheartj/eht394. 

8.     Picano E, Vano E. Updated Estimates of Radiation Risk for Cancer and Cardiovascular Disease: Implications for Cardiology Practice. J Clin Med. 2024 Apr 2;13(7):2066. doi: 10.3390/jcm13072066." 

Author Response

Comment 1: Page 2. The authors begin by discussing the safety of EST, which is a reasonable starting point for the review. They state that the myocardial infarction and death rate is approximately 1 in 2,500 ESTs. However, this statistic is drawn from Kharabsheh SM, Al-Sugair Abdulaziz, Al-Buraiki Jehad, and Farhan J.'s "Overview of Exercise Stress Testing" published in Annals of Saudi Medicine. This review, published in 2006, was based on even older and smaller data sets. In contrast, one of the largest multicenter studies published in the same year showed a much lower major adverse cardiac event (MACE) rate. This study, which included 26,295 ESTs, reported an event rate of 1 in 6,574 cases, with no deaths during the exercise tests [1]. Moreover, recent single-center studies have shown even lower complication rates. For example, [2] a study comprising 10,250 patients reported no deaths, acute myocardial infarctions, ventricular fibrillation, or asystole, with only non-sustained ventricular tachycardia occurring in 114 cases (1.1%). Modern studies also report no severe complications during exercise, likely due in part to the use of semi-supine exercise stress echocardiography. During modern exercise echocardiography, severe ischemia is visualized earlier, often before ST-changes or symptoms appear, resulting in shorter and safer tests. Therefore, I do not agree with the claim that the myocardial infarction and death rate is approximately 1 in 2,500 ESTs in 2024. This should be reflected in contemporary papers.

Response 1:
Thank you so much for pointing this out. We added “However, this was based on old data and recent studies showed almost no major complications occurring during stress echocardiogram, with the most frequent complication in one study is arrhythmias, most commonly Supraventricular extrasystoles” to the lines 82-85

Comment 2:
Page 1-2. Introduction/Indications for stress testing and further… Several times throughout the text, the authors mention in passing that the indication for EST is as a diagnostic tool for patients with chest pain. This is an outdated and narrow approach that limits the scope of EST's potential applications. In addition to chest pain, dyspnea is a very important symptom that can also serve as an indication for EST. Furthermore, there are many other pathologies that could warrant the use of EST [3,4].

Response 2: Thank you for pointing this out. We added “EST may be considered for patients with symptoms of typical or atypical angina pectoris who have low or intermediate pretest probability [68]” in the line 87-88 and "The recent guidelines from the European Society of Cardiology prefer stress imaging modalities over ECG EST, ECG EST can still be used for assessing exercise tolerance, hemodynamic response, and evaluating arrhythmias [68]." In lines 92-94

Comment 3: Page 5. The paper compares the rate of ST-depression during exercise with different APMHR formulas. However, this comparison is of limited importance since the diagnostic sensitivity of ST-depression is not very high, as noted in the new guidelines [4]. This is why imaging-based EST is now considered the first line of assessment for most patients. The study on differences in ST-segment depression was published in 2011 and is a pilot study involving 306 patients.

Response 3: Thank you for pointing this out. We have already added, “It was only abnormal in 28% of the individuals who exercised for >3min after reaching 85% of APMHR, which was attributed to post-test bias; however, a false positive test result cannot be totally excluded. Thus, the accuracy (false positive rate) of a more intense exercise versus a goal of 85% of the APMHR is unclear.”  We added “Therefore, the degree of ST-segment depression should not be used as a marker for the extent of induced ischemia during EST.” in lines 255-257

Comment 4: The paper is a review, not an original study, so modern modalities should be thoroughly described. However, there is only a single sentence in the “HRmax and Nuclear Imaging Studies” section regarding stress echocardiography. Stress echocardiography, by the way, can provide a number of new parameters with proven prognostic value. In relation to the paper's topic, heart rate reserve (HR reserve) has been proven to be an independent predictor of mortality [5]. It is a simple yet important parameter that can be used alone or in combination with other parameters, as demonstrated in the large multicenter study, SE2020.

Response 4: Thank you for pointing this out. We added “and aid in ischemia detection via the presence of regional wall motion abnormalities during the test [69]. Moreover, the heart rate reserve, calculated as peak HR/rest HR, is the 'E' criterion of the ABCDE protocol used in the assessment of CAD [69]. A heart rate reserve is considered positive if it is less than 1.80 during an exercise stress echocardiogram and it has HR of 2.955, (95% CI 1.848–4.725; P < 0.001) for all-cause mortality when positive [69]” in lines 374-379

Comment 5:   Page 8. In my opinion, the “HRmax and Nuclear Imaging Studies” paragraph should be significantly reworked. The authors analyze a 1989 study involving 330 patients, discussing the high and adequate ischemia detection using the SPECT thallium test. This might give readers the impression that this test is a good alternative for ischemia detection in current clinical practice. However, it should be emphasized that this test has the same accuracy as the older and simpler exercise stress echocardiography. Furthermore, safety concerns should also be highlighted. Since 2010, numerous studies have pointed out the significant dangers associated with SPECT tests.  For example, “Using radiation safely in cardiology: what imagers need to know. Heart. 2019: “Might best practices for SPECT myocardial perfusion imaging: 1. Avoid thallium stress…” [6].

“The appropriate and justified use of medical radiation in cardiovascular imaging: a position document of the ESC Associations of Cardiovascular Imaging, Percutaneous Cardiovascular Interventions and Electrophysiology, 2014” [7] etc., including recent papers [8]. This issue should be discussed meticulously and accurately.

Response 5: Thank you for pointing this out. We added “It is crucial to acknowledge that radiation exposure has numerous side effects, including damage to the skin, hair, and eye lens, as well as the potential for germ cell mutations and cancer development [69]. Contrary to the belief that only higher doses lead to cancer, there is no threshold level below which cancer cannot occur [70], meaning even small doses carry a risk. Stress echocardiograms with contrast have been found to be as accurate as 99-technetium SPECT in diagnosing coronary artery disease [71], hence it may be a better option to reduce the risk of radiation exposure” to the lines 399-405.

Round 2

Reviewer 3 Report

Comments and Suggestions for Authors Dear authors,   The manuscript has been improved in the comparison with the previous version. However, the more information about imaging exercise tests would be better. Check, please, the references carefully. They are mixed up.

Author Response

Comment 1: Dear authors,   The manuscript has been improved in the comparison with the previous version. However, the more information about imaging exercise tests would be better. Check, please, the references carefully. They are mixed up.

Response 1: Thank you so much for your comment. We adjusted the manuscript accordingly and stressed on the importance of stress echocardiography. Kindly find the changes in the lines 375-392.
